# Impact of General and Special Education Teachers' Knowledge on Their Practices of Self-Regulated Learning (SRL) in Secondary Schools in Riyadh, Kingdome of Saudi Arabia

**Huda Mutlaq Alenezy \*, Kee Jiar Yeo and Azlina Mohd Kosnin**

School of Education, University Technology Malaysia (UTM), Johor Bahru 81310, Malaysia;
kjyeo@utm.my (K.J.Y.); p-azlina@utm.my (A.M.K.)
\* Correspondence: h24.m@hotmail.com

**Abstract:** Teachers of students with disabilities have been assessed from various perspectives. This study aimed to investigate the knowledge and practice of self-regulated learning (SRL) of teachers who teach students with learning disabilities (SLD) in secondary schools in Riyadh, Saudi Arabia. The study focuses on the impact of teachers' knowledge in their practices of SRL and identified the moderating effect of teachers' gender on this impact. Using Pintirich's model of SRL, the study designed the Teachers' Knowledge of Self-Regulated Learning Scale, and it adopted Huh's Teachers' Practices Related to Developing Students' SRL questionnaire to examine teachers' knowledge and practices of SRL. The questionnaire was distributed online to over 318 Saudi teachers in secondary schools in Riyadh who were selected by using stratified sampling techniques. Teachers demonstrated high knowledge of SRL in all domains (cognition: $M = 5.2$, motivation: $M = 5.38$, behavior and emotions: $M = 5.44$) and medium SRL practices level in their classes ($M = 3.5$) with some reservation on their reaction and reflection, which were at the lowest average level. Furthermore, results revealed a direct and significant influence of teachers' knowledge on the practice of self-regulated learning (SRL) ($\beta = 0.183$, $t = 3.301$, $p = 0.000$), and there was no moderate effect of teachers' gender on the impact of teachers' SRL knowledge on teachers' SRL practices ($\beta = -0.004$, $t = 0.064$; $p = 0.949$). The results suggest the demand for more practical training programs for SLD teachers to increase their SRL application in practice.

**Keywords:** self-regulated learning (SRL); students with learning disabilities (SLD); special education; teachers (SE); general education teachers (GE)



## 1. Introduction

The concept of self-regulated learning (SRL) is distinct within the educational field. As posited by [1] (p. 1), "*SRL is a core conceptual framework to understand the cognitive, motivational, and emotional aspects of learning*". SRL is not just about individuals self-directing their learning to achieve their performance goals. While self-directed learners control the external learning environment, self-regulated learners focus is on internal factors such as cognition and motivation [2]. Zimmerman explained SRL as an efficient and independent learning that needs metacognitive skills, motivation, and strategic action, and he defined self-regulation as "*self-generated thoughts, feelings, and actions that are planned and cyclically adapted to the attainment of personal goals*" [3] (p. 14). Developing learners' SRL is a crucial requirement, as contemporary education requires independent learners who can manage the expanding knowledge, especially with the spread of technology and therefore the actual need for self-learning [4]. Furthermore, SRL is the foundation for ensuring lifelong learning [5–7].

Self-regulated learning (SRL) depends on the idea that learners must take responsibility for their own learning and participate actively in learning process [8]. It is derived from metacognition theories [9–11]. Zimmerman explained SRL in that "*Students can be described*

*as self-regulated to the degree that they are metacognitively,' motivationally, and behaviorally active participants in their own learning process*" [3] (p. 329). In addition, [12] (p. 453) argued the term of SRL as "*An active, constructive process whereby learners set goals for their learning and then attempt to monitor, regulate, and control their cognition, motivation, and behavior, guided and constrained by their goals and the contextual features in the environment*". Most definitions agree that self-regulation is a process based on predetermined goals. The current study adopts Pintrich's definition since it distinguishes between the conceptual aspect between SRL and metacognition, and it takes into account the empirical association with motivation.

### 1.1. Background

In the Kingdom of Saudi Arabia (KSA), there is limited evidence of teachers' knowledge and practice of SRL in both general classes and resource rooms where SLD receive their individual educational plan (IEP). Despite numerous research on teachers' SRL knowledge and practice, little is known about teachers' SLD knowledge and practice of SRL in other countries or in Saudi Arabia. Furthermore, the majority of these studies focused on primary school teachers, with only a few studies looking into secondary school teachers' knowledge and practice of SRL [13]. The moderating effects of teachers' gender between teachers' knowledge and practice of SRL also remain unclear and underexplored in this context.

Therefore, this study aimed to investigate the general education (GE) and special education (SE) teachers' knowledge on SRL and practice of SRL. It was also aimed at identifying the impact of teachers' knowledge in their practices of SRL and the moderating effect of gender on the relationship between teachers' knowledge on SRL and their practices of SRL.

### 1.2. Self-Regulated Learning (SRL) and Students with Learning Disabilities (SLD)

SRL has been shown to improve students' academic performance, motivation, and social and cognitive skills [14–17]. Therefore, teaching self-regulation skills is critical in both general and special education settings [18] as it benefits both students with learning disabilities (SLD) and their peers [19–22]. Students with learning disabilities (SLD) are one of the various categories of students. Learning disabilities (LD) are defined as a having lack of specific cognitive processes and inadequate academic accomplishment while having a normal or above-average IQ [23]. LD is one of the most common types of special needs [24,25]. SLD account for over half of all students with special needs (SSN) in the Kingdom of Saudi Arabia (KSA), accounting for 42 percent of all SSN [26].

The features of learning disabilities vary according to age. For example, SLD in secondary school fail to achieve course requirements when compared to their typically developing peers [27]. Their difficulties can be shown in their study skills, which include taking notes, comprehending instructional materials, and paying attention and focusing. Hence, in order to achieve success, SLD must understand the conditions of success as well as how to learn more effectively [28]. More attention should be paid to promote SRL among students, as studies have demonstrated its positive impact on SLD and their typically developing peers' academic, psychological, and social performance [19,21,22,29,30].

### 1.3. Influence of Self-Regulated Learning on Students with Learning Disabilities (SLD)

Students with learning disabilities (SLD) have challenges in planning study time and developing skills. They often lack academic, personal, and social skills that students must acquire, such as self-regulation skills: self-evaluation, understanding of one's own strengths and needs, self-advocacy, and self-restraint (Hong et al., 2007; Wehmeyer, 1996, as cited in [31]). In this regard, SLD students must acquire skills that can help them to become self-reliant in terms of obtaining knowledge and improving abilities, such as self-regulated learning (SRL) skills. Btonutler and Schnellert [32] emphasized the significance of developing SRL for SLD.

The main goal of self-regulated learning is to enhance learning tasks and performance by reconstructing the learners' efforts with instructional strategies underlying the cognitive, metacognitive, affective, and motivation processes [33]. The main influence of

self-regulated learning appears to be academic performance of students or the academic productivity. Rohrkemper [34] argued the relationship between self-regulation and academic performance since the student's usage of self-regulated strategies as an adaptive learning ability led to higher grades and high-quality profiles compared to those less-adaptive learners. Lucieer et al. [35] also confirmed the positive relationship between the academic performance and self-regulation skills (such as planning, monitoring, evaluation, reflection, effort, and self-efficacy) of students. Additionally, [36] added that there is a statistically significant relationship between SRL skills (such as motivation, setting learning goals, learnings strategies, and self-monitoring) and learning performance. Broadbent, and Poon, [37] also found a positive association between self-regulated learning and students' achievement in both offline and online teaching modes by conducting a systematic review of twelve studies over a period of 14 years.

On the other hand, multiple studies such as [38–40] argued the impact of SRL skills in increasing students' self-efficacy and progression in learning. Kennedy and Krause [41] emphasized that SLD who demonstrate a higher level of self-efficacy, set goals, and employ efficient strategies are more likely to achieve academically in college. Social cognitive theory explains self-efficacy as a context-specific variable influenced by using SRL strategies to enhance the interaction between personal, behavioral, and environmental aspects. Furthermore, the age of student was discussed as an influenced variable when studying the relationship between SRL and self-efficacy by multiple researchers, including [38], who approved the strong relationship in 9- to 10-year-old students. Li and Zheng [39] approved the same relationship in 11- to 16-year-old students, and [40,42] approved the strong relationship at 16 to 17 years old. Higher education levels also demonstrated the strong relationship [43].

Among the skills to be a self-regulated learner are complex cognitive processes. There also exists congruent empirical evidence of the effectiveness of self-regulated learning for students with learning disabilities (SLD) to enhance their abilities in problem solving [19,44,45] and to improve SLDs' language (comprehension, writing, and so forth) [46,47]. Moriña and Biagiotti [48] confirmed that self-regulation as a part of executive functioning is an essential academic success factor for SLD in university that allows them to face everyday problems easily, set their goals, and adapt to their environments.

*1.4. Role of Teachers in Self-Regulated Learning (SRL) Practices among Students with Learning Disabilities (SLD)*

According to Bandura's social cognitive theory, SRL does not happen by itself [49]. This happens with SLD, as SRL does not emerge naturally [50]. This means that students require a role model in order to learn these skills; as a result, teachers play an important role in promoting and training SRL skills [3,51–55]. In fact, even if teachers do not explicitly teach SRL, some of them may provide opportunity for students to practice it [53].

Teachers generally deliver SRL to their students in both direct and indirect methods. Direct methods to promote SRL are classified into two types: implicit and explicit. The implicit way occurs when a teacher encourages their students to engage in strategic behavior without teaching the strategy or when a teacher applies the strategy in front of students without explaining its steps and importance [56]. On the other hand, explicit instruction happens when teachers explain to pupils why, how, and when to adopt a given strategy [57,58]. Besides the direct method, teachers can promote SRL by creating an appropriate learning environment in which students can actively participate in their learning [58].

Many factors can influence teachers' practice to promote SRL among students, including their knowledge and self-efficacy [59,60]. However, adequate knowledge of SRL does not always imply that teachers will use it directly [61] since teachers' knowledge can have an indirect effect on their promotion of SRL via teacher self-efficacy [60]. Furthermore, gender may have an impact on the relationship between teachers' SRL knowledge and their practice [58].

According to [62], students learn more SRL skills from intervention programs given by researchers than they do from teachers. That can be attributed to the lack of teachers' knowledge on SRL [63] or their lack of SRL instruction in the classroom [57]. Additionally, [64] explained the importance of using intervention projects in schools for SLD to (i) increase the consistent engagement of SLD in interventions in the class, (ii) increase the results and expected academic outcomes, and (iii) enhance students' independence and self-direction, self-confidence, pride, and sense of control over learning and understand the significance of their individualized learning strategies in terms of their academic performance.

Several studies have investigated the role of teachers' knowledge in their classroom practices in general, as [65] confirmed: the teacher competency is dependent on content knowledge linked to the subject and pedagogical content knowledge related to teaching strategies and practices to promote subject accessibility. In addition, [66] interpreted the relationship between teachers' knowledge and their performance in the classroom since the limitation of knowledge is an influential factor on teachers' performance in illustrating, selecting tasks, drawing connections, and selecting and adapting instructional strategies.

SRL is one of the practices that teachers must have to instruct to students [67–71]. This is because the teachers' knowledge about SRL instruction as content knowledge and pedagogical content knowledge about SRL is related positively with SRL implementation and explained variance in implementation levels [72]. Spruce and Bol [70] added that even though teachers showed a medium level of SRL knowledge and a low level of SRL practices in the classroom, the teachers who had higher knowledge exhibited the strongest implementation of SRL in the classroom.

Ewijk and Werf [73] also added that most teachers' knowledge about SRL was limited to the characteristics of constructivist learning environments, while a few of them mentioned teaching strategies as part of the SRL. In addition, [60] reported a limited extent of SRL instruction in classrooms. In contrast, [74] (p. 195) reported high trends of SRL practices in classrooms, such as "goal setting, modelling, scaffolding, and developing learner autonomy". However, problem-solving and critical thinking practices were not promoted.

On the other hand, the degree of SRL practices among teachers varies in learning environments since it is not always the same as or even related to classroom practice observation results by the same teachers [56,72]. Even though most teachers encouraged SRL during learning and intended to implement SRL, they could not do it due to various reasons such as lack of knowledge, competencies, or contextual factors [72,75].

Based on that, the teachers use their own private and unique strategies to support SRL in the classroom, such as "promote social interaction, transforming students from individualization to socialization, mediations, directing from simple to complex processes, reflections, evaluations of learning, and moving from social interactions to SRL" [76] (p. 5). In addition, some of the teachers promote cognitive and motivational parts of SRL more than meta-cognition and strategic actions [76]. Ewijk et al. [56] also added the teachers have a strong inclination to use implicit SRL strategy in the classroom by modeling a certain method without providing explicit instruction.

The impact of teachers' gender on their practice of SRL was discussed by [58]. It was found that female teachers were keener to try the instruction SRL methods than their male counterparts. This result is ascribed to traditional social roles that claim that male teachers are more inclined to perform lecturing and authorizing. However, [50,63,77–79] confirmed that gender has no significant impact on SRL and its promotion.

## 2. Methods and Materials

### 2.1. Research Design and Participants

Quantitative cross-sectional research design was employed in this study. It is a descriptive analytical study using questionnaires via online platform to obtain responses from participants on their knowledge about SRL and practices of SRL.

Participants in this study are teachers from Riyadh city, which has the largest population of teachers in Saudi Arabia [80]. The study involved a total of 318 general education

and special education teachers from all secondary schools that include a learning disabilities program in Riyadh for the academic year 2021/2022. Participants differ in terms of gender and specializations.

Stratified sampling procedures were used to determine the effect size of the samples (participants). Specifically, the disproportionate allocation for within-strata analysis was used because there is a large discrepancy between the number of female and male teachers. There was also a large discrepancy between the number of general education teachers (GE) and special education (SE) teachers, where the former is much more than the latter. According to [81], disproportionate allocation in within-strata analysis may be more appropriate than proportionate stratification, as the sample size in some strata is very small. As shown in Table 1, from the total number participants, 53 were SE teachers (13 male and 40 female teachers), and 265 were GE teachers (73 and 192 female teachers).

**Table 1.** Participants' demographic data.

| Variable | Categories | Frequency | Percent |
|---|---|---|---|
| Specialization | General Education Teacher | 265 | 83.3 |
| | Special Education Teacher | 53 | 16.7 |
| Gender | Male | 86 | 27.0 |
| | Female | 232 | 73.0 |
| School Type | Public School | 272 | 85.5 |
| | Private School | 46 | 14.5 |
| Years of Experience | 1–5 Years | 31 | 9.7 |
| | 6–10 Years | 77 | 24.2 |
| | 11–15 Years | 86 | 27.0 |
| | Over 15 Years | 124 | 39.0 |

*2.2. Measures*

2.2.1. Instruments

Responses from participants were obtained through a questionnaire that contains three sections. The first section contains items on teachers' demographics: gender, specialization, years of experience, and school type. The second section contains items on teachers' knowledge on SRL by using the Teachers' Knowledge Self-Regulated Learning Scale, developed by the researchers based on Pintrich's Model (2000) [82]. This section used a six-point Likert scale (1 = strongly disagree, 2 = disagree, 3 = slightly disagree, 4 = slightly agree, 5 = agree, and 6 = strongly agree). The third section featured items on teacher practices to reinforce SRL in classrooms using the Teachers' Practices Related to Developing Students' SRL Survey, developed by [83]. This section used a five-point Likert scale (1 = never, 2 = rarely, 3 = not sure, 4 = sometimes, 5 = always). (See Appendix A)

The questionnaire was designed in Google Form and sent via an electronic link to teachers by school principals. It was not possible to meet the teachers face to face due to the conditions of COVID-19. Instructions were placed at the beginning of the questionnaire with a contact number for any inquiries from participants.

To ensure the validity of the questionnaire, experts' validation and exploratory and confirmatory factor analysis to calculate the psychometric properties of measurement tools were used. In addition to validation, the reliability of the instrument was also obtained by way of a pilot study, with results showing that the alpha coefficient exceeded 0.80 for each subscale An alpha coefficient of 0.80 is considered an acceptable level of internal reliability [84], while others suggested that 0.67 or above is an acceptable reliability level [85].

2.2.2. Data Collection

Data collection took around three months. Before starting to collect data, lists of the names of secondary schools that have learning disabilities programs in Riyadh and the number of GE teachers and SE teachers in these schools were obtained from the General

Administration of Education in the Riyadh region. This was followed by approval to implement the research in these schools.

There were 41 secondary schools that have a learning disabilities program in Riyadh, of which 11 schools were for male students, while 30 were for female students. The aims of the study and the required samples were explained to schools' principals. Each school principal was asked to choose at least 10 GE teachers who teach SLD in general classes and all SE teachers in their school to answer the questionnaire. An electronic link containing the questionnaire was sent to the principals via WhatsApp, and it was later shared with their teachers.

### 2.2.3. Data Analysis

Means and standard deviations by SPSS 25 were used to answer the first question that related to the levels of knowledge and practices of self-regulated learning among SE teachers and GE teachers. In order to analyze the research model, partial least squares (PLS) was utilized, while the Smart PLS 3.0 software was used as the analysis technique [86]. The measurement model (validity and reliability of the measures) was investigated based on the recommendation of two-stage analytical procedures by [87], then followed by an evaluation of the structural model testing the hypothesized relationships [88,89]. In order to check the significance of the loadings and the path coefficients, a bootstrapping method was used [89]. Finally, to investigate the role of teachers' gender as a moderator variable in the impact of teachers' knowledge and practices of SRL, the product indicator approach by Smart PLS software was conducted.

### 2.3. Ethics

Prior to distributing the questionaries, the researchers contacted Dr. Yeol Huh via e-mail to obtain his permission to use his tool, Teachers' Practices Related to Developing Students' SRL Survey. Psychometric properties were obtained to ensure quality of the results.

All participants gave their informed consent before their access to the questionnaire. Participants were informed that their participation in this questionnaire would be kept confidential, their anonymity was assured, and data would be used for research purposes only.

## 3. Results and Findings

### 3.1. Levels of Knowledge and Practice of Self-Regulated Learning (SRL) among Secondary General Education (GE) and Special Education (SE) Teachers

#### 3.1.1. Levels of Knowledge of SRL among Secondary GE and SE Teachers

The results showed that most of the teachers who participated in this study had high knowledge of SRL in all its domains (cognition, motivation and behavior, and emotions). The highest mean value for behavior and emotions was 5.44, followed by motivation with an average value of 5.39, while cognition ranked third with an average value of 5.27 (Table 2).

**Table 2.** Teachers' Knowledge on Self-Regulated Learning (SRL).

| Knowledge Domain | Skill | Code | Min | Max | M | SD | Level |
|---|---|---|---|---|---|---|---|
| Cognition | Planning | KCP | 3.67 | 6.00 | 5.27 | 0.485 | 5.27 High |
| | Self-Regulation | KCSR | 3.17 | 6.00 | 5.25 | 0.568 | |
| | Self-Evaluation | KCSE | 2.50 | 6.00 | 5.30 | 0.585 | |
| Motivation | Planning | KMP | 4.00 | 6.00 | 5.42 | 0.514 | 5.39 High |
| | Self-Regulation | KMSR | 3.40 | 6.00 | 5.48 | 0.486 | |
| | Self-Evaluation | KMSE | 2.00 | 6.00 | 5.27 | 0.604 | |
| Behavior and Emotion | Planning | KBP | 3.33 | 6.00 | 5.52 | 0.510 | 5.44 High |
| | Self-Regulation | KBSR | 2.00 | 6.00 | 5.47 | 0.632 | |
| | Self-Evaluation | KBSE | 3.00 | 6.00 | 5.33 | 0.611 | |

Note: M, mean; SD, standard deviation; Min, minimum; Max, maximum.

### 3.1.2. Levels of Practice of SRL among Secondary GE and SE Teachers

The results revealed a medium practice level for SRL among teachers participating in this study. The monitoring phase had the highest mean value of 3.59, followed by forethought, planning, and activation with the second highest value with an average of 3.57. Control came in third with an average value of 3.51, while reaction and reflection showed the lowest average value (3.48) (Table 3).

**Table 3.** Teachers' practice of self-regulated learning SRL.

| Dimension | Code | Min | Max | M | SD | Level |
|-----------|------|-----|-----|---|----|-------|
| Forethought, planning, and activation | PF | 1.00 | 5.00 | 3.57 | 0.949 | |
| Monitoring | PM | 1.00 | 5.00 | 3.59 | 1.016 | 3.54 |
| Control | PC | 1.00 | 5.00 | 3.51 | 0.952 | |
| Reaction and reflection | PR | 1.00 | 5.00 | 3.48 | 0.979 | |

Note: M, mean; SD, standard deviation; Min, minimum; Max, maximum.

### 3.2. Impact of Secondary General Education (GE) and Special Education (SE) Teachers' Knowledge on Their Practice of Self-Regulated Learning (SRL)

The impact of teachers' knowledge on their practices of self-regulated learning (SRL) in secondary schools was measured by correlation and moderating effect. Before the analysis, the measuring model was determined to ensure the validity and reliability of all the responses from 318 teachers.

### 3.2.1. Assessment of the Measurement Model

The relationship between knowledge and practice on self-regulated learning (SRL) among teachers of students with learning disabilities (SLD) was tested using partial least square (PLS) method. Analysis of the measurement model (or outer model), the first step of PLS analysis, was used to determine the appropriateness of the theoretically defined construct. The measurement model was examined to ensure the survey questionnaire determines the variables that were supposed to be measured and to simultaneously make sure that the instrument is reliable. In this process, three things were investigated, which are factor loadings, composite reliability (CR), and average variance extracted (AVE) (Figure 1).

According to [90], the construct validity and reliability can be accessed via investigating the loadings, average variance extracted (AVE), and composite reliability (CR), where [90] suggested that the loadings should be > 0.60, AVE > 0.5, and CR > 0.7. As shown in Table 3, the AVE was greater than 0.5, and the CR was greater than 0.7. Hence, after deleting all the problematic items, all loadings retained in the model were above the cut-off value (0.60). AVE and CR values were within the recommended values > 0.50 and >0.70, respectively. Therefore, it can be concluded that the measures used in the present study show appropriate convergent validity and reliability (Table 4).

Furthermore, the discriminant validity was assessed based on Fornell and Larcker [91] prescription, where the average variance extracted (AVE) are compared with squared correlations, or alternatively, the square root of the AVE are compared with the correlations. As indicated in Table 4, the square roots of the AVE (bolded) are all more than the off-diagonal correlation values, suggesting that there is sufficient discriminant validity. Therefore, it can be concluded that the measures used in the present study show appropriate divergent validity. In general, the findings indicate that the construct validity for this study is reliable, and the model has psychometrically sound properties (Table 5).

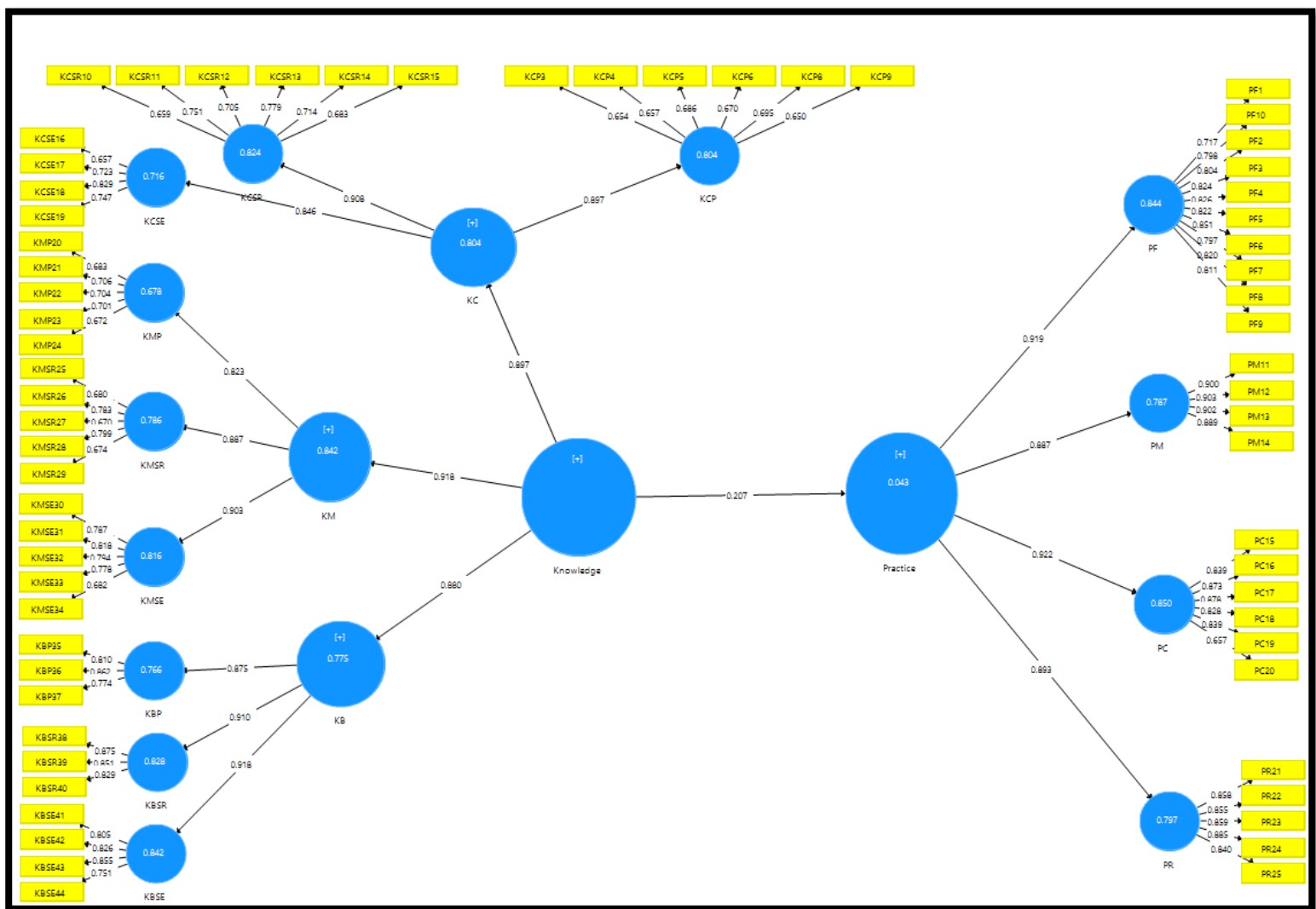

**Figure 1.** The Measurement Model.

**Table 4.** Results for the Measurement Model.

| Variable | Dimension | Constructs | (AVE) (>0.5) | Composite Reliability (>0.7) |
|---|---|---|---|---|
| Behavior and Emotions | Planning | KBP | 0.666 | 0.857 |
| | Self-Evaluation | KBSE | 0.656 | 0.884 |
| | Self-Regulation | KBSR | 0.725 | 0.888 |
| Cognition | Planning | KCP | 0.548 | 0.829 |
| | Self-Evaluation | KCSE | 0.550 | 0.829 |
| | Self-Regulation | KCSR | 0.513 | 0.863 |
| Motivation | Planning | KMP | 0.501 | 0.822 |
| | Self-Evaluation | KMSE | 0.598 | 0.881 |
| | Self-Regulation | KMSR | 0.524 | 0.845 |
| Practice | Control | PC | 0.676 | 0.926 |
| | Forethought, planning, and activation | PF | 0.652 | 0.949 |
| | Monitoring | PM | 0.807 | 0.944 |
| | Reaction and reflection | PR | 0.739 | 0.934 |

**Table 5.** Discriminant Validity of Measurement Model.

| | KBP | KBSE | KBSR | KCP | KCSE | KCSR | KMP | KMSE | KMSR | PC | PF | PM | PR |
|---|---|---|---|---|---|---|---|---|---|---|---|---|---|
| KBP | **0.816** | | | | | | | | | | | | |
| KBSE | 0.689 | **0.81** | | | | | | | | | | | |
| KBSR | 0.727 | 0.745 | **0.852** | | | | | | | | | | |
| KCP | 0.473 | 0.563 | 0.52 | **0.669** | | | | | | | | | |
| KCSE | 0.458 | 0.584 | 0.524 | 0.652 | **0.742** | | | | | | | | |
| KCSR | 0.526 | 0.575 | 0.516 | 0.73 | 0.668 | **0.716** | | | | | | | |
| KMP | 0.511 | 0.533 | 0.499 | 0.485 | 0.537 | 0.579 | **0.693** | | | | | | |
| KMSE | 0.59 | 0.686 | 0.636 | 0.565 | 0.637 | 0.628 | 0.606 | **0.773** | | | | | |
| KMSR | 0.596 | 0.599 | 0.577 | 0.483 | 0.586 | 0.542 | 0.613 | 0.704 | **0.724** | | | | |
| PC | 0.051 | 0.103 | 0.142 | 0.133 | 0.16 | 0.173 | 0.16 | 0.1 | 0.076 | **0.822** | | | |
| PF | 0.11 | 0.205 | 0.187 | 0.194 | 0.218 | 0.206 | 0.165 | 0.138 | 0.137 | 0.745 | **0.808** | | |
| PM | 0.1 | 0.109 | 0.132 | 0.155 | 0.174 | 0.143 | 0.107 | 0.092 | 0.087 | 0.795 | 0.758 | **0.899** | |
| PR | 0.064 | 0.137 | 0.166 | 0.151 | 0.226 | 0.186 | 0.158 | 0.138 | 0.145 | 0.848 | 0.716 | 0.73 | **0.86** |

Note: (i) Shown bolded are the square of the average variance extracted (AVEs) for each construct; (ii) the squared inter-factor correlation values, which are also known as shared variance, are presented below the diagonal of the table.

### 3.2.2. Structural Model to Measure the Relationship between GE and SE Teachers' Knowledge and Practices of SRL

The next step was assessment of the structural hypothesized model (inner model) using SEM analysis to examine the hypothesized relationships between knowledge and practice on self-regulated learning (SRL) among the teachers who teach students with learning disabilities (SLD). First, the weights or path coefficients of the relationships were investigated and tested for their significance through t-values obtained from the bootstrapping method. The result shows that teachers' knowledge on SRL directly and significantly influence practice on self-regulated learning (SRL) (β = 0.183, t = 3.301, *p* = 0.000). Moreover, the coefficient of determination, $R^2$ for dependent variables, was assessed in order to find the amount of variance in each construct, which is described by the model. The testing of the significance for the regression weights are achieved by running 1000 bootstrapped samples from the original 318 cases (Figure 2).

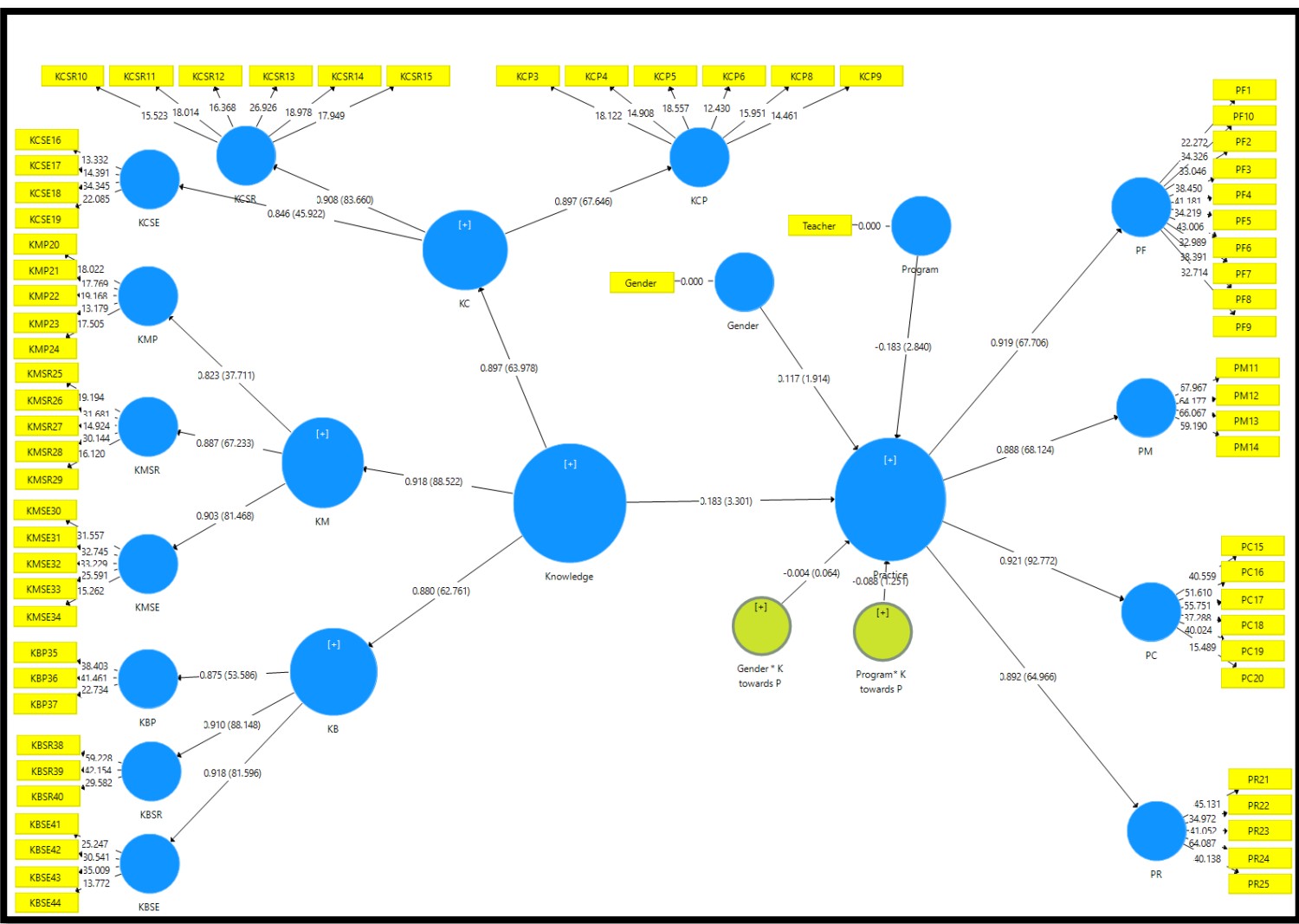

**Figure 2.** The Structural Model.

*3.3. The Moderating Effect of Secondary General Education (GE) and Special Education (SE) Teachers' Gender in the Impact of Their Knowledge on Their Practices of Self-Regulated Learning*

The product indicator approach, proposed by Chin (1996), was conducted to examine the role of teachers' gender as a moderator variable in the impact of teachers' knowledge and practices of SRL [92]. To assess the moderation effects, Smart PLS software was used, particularly the product indictor approach. As shown in Table 6 ($\beta = -0.004$ t = 0.064; *p* = 0.949), the *p*-value is not significant, which provides evidence that teachers' gender does not moderate the impact of teachers' knowledge on practices of SRL (Table 6).

**Table 6.** Moderating effect of teachers' gender on the impact of their knowledge on their practices of SRL.

| Impact | Std. Beta | Std. Error | t-Value | *p*-Value | Decision |
|---|---|---|---|---|---|
| Gender K -> *p* | −0.004 | 0.06 | 0.064 | 0.949 | Not supported |

## 4. Discussion

The purpose of the current study is to associate SLD teachers' knowledge about SRL and their classroom practices. First, the study revealed a high level of SRL knowledge among SLD teachers in both general and special education streams. Teachers demonstrated a high-level knowledge in various components of knowledge cognition, motivation, and behavior and emotion of SRL. The behavior and emotion component was the highest component level of the high SRL knowledge components associated with the classroom practice observations that revealed that teachers are more likely to promote cognitive and motivational parts of SRL than meta-cognition and strategic actions [76].

The high level of knowledge may be ascribed to the Ministry of Education's emphasis in recent years on including new methods in university and college teacher preparation programs, pre-service programs, and in-service professional development. It is worth mentioning that collaborative discussion between SLD teachers and the SLD teachers' community contributes to increasing the opportunities for teachers to expand their knowledge about SRL and their personal sense of SRL development and develop a common explicit discourse of SRL in their practices. Furthermore, a high knowledge level can be attributed to teachers' responsibility for their learning and their beliefs that they can practice what they preach [68]. The literature asserted that teachers easily instruct and explain self-regulating learning (SRL) strategies and processes to their students if they were self-regulated themselves [67–71]. High knowledge of SRL means that teachers are more capable of recognizing opportunities in classrooms to foster SRL among students.

The results showed a medium level of SRL practicing in classrooms, with the lowest medium practice of reaction and reflection. It highlights the lack of teachers' ability to apply multiple and numerous strategies and activities to involve students in self-reflection. This finding suggests the demand for professional development to increase and expand teachers' understanding and practicing of various SRL-based instruction, explicit strategies, and implicit strategies in real classroom settings. Despite the medium level of practice of SRL in the classroom, they perceive a high-level knowledge of SRL. This aligned with [72,75], who disclosed that although teachers reported the intention to implement SRL, they could not do it due to various reasons such as lack of knowledge, competencies, or contextual factors [72]. Hence, there are other factors that may affect teachers' practices of SRL rather than knowledge, such as the availability of facilities, external resources, environmental setting, or policy constraints. On the other hand, teachers may have knowledge of a specific constructivism learning environment and not the SRL, which affects their practices of SRL. This is aligned with the findings of [70], which indicated that teachers say one thing yet do another, which means that while teachers may know what SRL is, they do not know how to do it or implement it. Since the current study did not address content and pedagogic knowledge level, the medium level of practice can be attributed to the lack of pedagogic knowledge.

The finding emphasizes the relationship between teachers' knowledge and their practices of SRL in the classroom. In general, teachers cannot teach what they do not know [68]. Having knowledge means demonstrating, illustrating, selecting tasks, drawing connections, and selecting and adapting instructional strategies [66]. Spruce and Bol [70] demonstrated higher-knowledge teachers exhibit the strongest implementation of SRL in the classroom. These findings support the fundamental significance of knowledge about SRL in SRL instructing and fostering among students reported by [67–71]. The result also agrees with Karlen et al. [72]'s outcomes that pedagogical content knowledge about SRL and content knowledge about SRL were statistically significantly related to self-reported SRL implementation and explained variance in implementation levels. The study findings also defended previous results of [50,63,77–79], which confirmed that gender does not significantly moderate the impact of teachers' knowledge and practices of SRL. In contrast, [58] asserted that the relationship between female teachers' knowledge and practices differs from their male counterparts and thus indicates a significant moderate impact of gender on the SRL knowledge impact on SRL practices.

## 5. Conclusions

The significance of instructing self-regulated learning for students with learning disabilities (SLD) makes it a priority because it can help them to become self-reliant in terms of obtaining knowledge and improving abilities. This is because, for some students such as SLD, SRL may not emerge naturally. Accordingly, teachers represent an opportunity to enhance self-regulation learning skills among SLDs. Thus, current study aimed to examine the teachers' knowledge about SRL and their classroom practices to develop SRL among students. Furthermore, the study investigated the association between SRL knowledge level and SRL practices level and if gender moderates the impact of SRL knowledge on SRL practices. The study was a descriptive analytical study that used an online questionnaire that assessed teachers' knowledge about SRL and teachers' practices of SRL. The study recruited 318 SLD teachers from general education and special education mainstreams in Riyadh using stratified sampling procedures. The study revealed that SLD teachers have acquired a high level of knowledge about SRL, but they rarely practice SRL in their classrooms to develop their student's SRL skills. There was a statistically significant impact of SRL knowledge on practices of SRL, and teachers' gender does not moderate statistically the impact of teachers' knowledge on practices of SRL. The current findings contribute to improving teachers' training in practicing and instructing SRL. Due to the fair level of the SRL practices in the classroom in both general and special education settings, intervention programs are recommended that are carefully designed to enhance teachers' practices of SRL since teachers have a high level of theoretical knowledge, but it is not adequate to improve SRL practices. Hence, a need has emerged for conducting empirical findings on the effectiveness of interventions programs to increase SLD teachers' practices in classrooms.

## 6. Limitation

As with all quantitative research, the current study has generality concerns. The number of targeted teachers was adequate, but volunteer bias could affect the generalizable results. Teachers may also respond differently because they believed that they will be assessed using their responses, or they answered in a socially desirable manner. Besides this, the characteristics of the education system and teacher training in KSA may also affect the generalizability of the results. Therefore, we encourage further studies in different geographical areas. In addition, this study is limited to the main purpose of investigating the impact of SRL knowledge level among SLD teachers and their SRL practices. The current study did not address differences in SRL knowledge level or SRL practices in the classroom due to various background information. Furthermore, although the study inherited self-administrated responses of knowledge and practices, the current study did not address whether the teachers' practices in the classroom are a purposive practice to develop SRL strategies and skills for their students or if it is implicated behavior rather than

explicit in norms that served in managing students' behavior. Threat effects on the results are another limitation since the study is only based on a questionnaire, and its reliability was conducted in a pilot study. For enhancing the trustworthiness of findings, mixed-method approaches are recommended that use different data resources that validate and increase the trustworthiness of findings. The study is also limited to the knowledge level and practices level of SRL and did not address the challenges behind both. Thus, future studies are encouraged to reveal the reason and challenges for SRL knowledge and SRL practices of SLD teachers.

**Author Contributions:** Conceptualization, H.M.A. and K.J.Y.; methodology, H.M.A.; software, H.M.A.; validation, H.M.A., K.J.Y. and A.M.K.; formal analysis, H.M.A.; investigation, H.M.A.; resources, H.M.A. and K.J.Y.; data curation, H.M.A.; writing—original draft preparation, H.M.A.; writing—review and editing, H.M.A. and K.J.Y.; visualization, H.M.A.; supervision, K.J.Y. and A.M.K. All authors have read and agreed to the published version of the manuscript.

**Funding:** This research did not receive any external funding.

**Institutional Review Board Statement:** Not applicable.

**Informed Consent Statement:** Informed consent was obtained from all subjects involved in this study.

**Data Availability Statement:** Data sharing not applicable.

**Acknowledgments:** The authors would like to acknowledge the contribution and the cooperation given by the respondents and institutions involved in this study.

**Conflicts of Interest:** The authors declare no conflict of interest.

### Appendix A

Teachers' knowledge and practices to promote self-regulated learning questionnaire
*Dear teacher,*

*The purpose of this study is to determine teachers' knowledge and their practices on self-regulated learning (SRL) among secondary students with learning disabilities in Riyadh.*

*This questionnaire consisted of 3 sections; the first section covers demographic data, while the following sections consist of teachers' knowledge on self-regulated learning (SRL) and teachers' practices of SRL.*

*If you are a general education or learning disabilities teacher in one of the secondary schools that apply a learning disabilities program in Riyadh, please participate in this questionnaire, noting that your participation is voluntary. Do not write down your name on this questionnaire. Your response will be anonymous and kept confidential for this study only.*

*Thank you for your cooperation.*

*Demographic data*

*Please circle the most appropriate response.*

(1) Name: (Optional):
  (a) General Education Teacher: Major
  (b) Special Education Teacher

(2) Gender
  (a) Male
  (b) Female

(3) I am working in:
  (a) Public School
  (b) Private School

(4) Number of years of teaching experience:
  (a) 1–5 Years
  (b) 6–10 Years
  (c) 11–15 Years
  (d) Over 15 Years

(1)  **Teachers' Knowledge on Self-Regulated Learning**

The following items identify your knowledge about self-regulated learning and how students can apply it. Please choose the most appropriate scale which represents your opinion. "Strongly agree" reflects what the self- regulated student should do, while "strongly disagree" reflects that you do not agree that the statement represents what the self-regulated student should do.

| Domain | Skills | Items | Strongly Agree | Agree | Slightly Agree | Slightly Disagree | Disagree | Strongly Disagree |
|---|---|---|---|---|---|---|---|---|
| Cognition | Planning | The student determines what s/he wants to learn (his/her goals) at the beginning of the task | | | | | | |
| | | The student must learn how to set his goals according to the time available | | | | | | |
| | | The student sets a plan for the activity he wants to do | | | | | | |
| | | The student must learn how to choose the right strategy to achieve the goal of the task | | | | | | |
| | | The student must be able to prioritize according to his strengths and weaknesses | | | | | | |
| | | The student should determine that the strategy he/she is using will be effective | | | | | | |
| | | The student must learn when to ask others for help to achieve the goal of the task before beginning the task | | | | | | |
| | | The student should possess ways to acquire certain skills related to learning | | | | | | |
| | | The student should ask him/herself questions about the task before starting it | | | | | | |
| | Self-regulation | The student should pause while performing the task to check his progress towards completion | | | | | | |
| | | The student must monitor his understanding while performing the task | | | | | | |
| | | The student should check his time management proficiency while performing a task | | | | | | |
| | | The student must check his performance by analyzing his errors while performing the task | | | | | | |
| | | The student should take self-notes when performing the task | | | | | | |
| | | The students must change the used strategy during the task if he discovered that it is insufficient to achieve specified goals | | | | | | |
| | Self-evaluation | The student should verify the achievement of his goals in light of the available time and resources by comparing his answer with the model answer | | | | | | |
| | | At the end of the task, the student must evaluate the strategy used in terms of success and failure | | | | | | |
| | | At the end of the assignment, the student should evaluate his level of satisfaction with performance outcomes | | | | | | |
| | | After completing the assignment, the student should summarize what he has learned | | | | | | |

| Domain | Skills | Statement | Strongly agree | Agree | Slightly agree | Slightly disagree | Disagree | Strongly disagree |
|---|---|---|---|---|---|---|---|---|
| Motivation | Planning | The student must get rid of the distractions around him | | | | | | |
| | | The student must change his surroundings if he needs to be able to concentrate on work | | | | | | |
| | | The student should select a place that he would like to work in | | | | | | |
| | | The student must divide the task into sub-tasks | | | | | | |
| | | The student should set sub-goals for the task | | | | | | |
| | Self-regulation | The student must reward himself when he performed correctly during the task | | | | | | |
| | | The student should work in a way that makes the task more pleasant | | | | | | |
| | | The student should relate his study task to his real life | | | | | | |
| | | The student must believe in the importance of good performance in lessons and exams | | | | | | |
| | | The student must work harder to obtain good grades | | | | | | |
| | Self-evaluation | The student should revise his persistence level to achieve his goals | | | | | | |
| | | The student should revise the level of challenges he faced during completing the task | | | | | | |
| | | The student should revise his interest level as an ongoing task | | | | | | |
| | | The student should determine how to enhance his persistence and interest in an ongoing task | | | | | | |
| | | The student must use the self-reward upon successful completion of each step of the task | | | | | | |

| Domain | Skills | Statement | Strongly agree | Agree | Slightly agree | Slightly disagree | Disagree | Strongly disagree |
|---|---|---|---|---|---|---|---|---|
| Behavior and emotions | Planning | The student should learn how to control his emotions and behaviors | | | | | | |
| | | The student should learn how to be ready to start his school day actively. | | | | | | |
| | | The student should plan his behavior in different situations | | | | | | |
| | Self-regulation | The student should observe his emotions during the situation | | | | | | |
| | | The student must control his emotions in different situations | | | | | | |
| | | Students must use effective strategies such as deep breathing to control his emotions | | | | | | |
| | Self-evaluation | The student should reconsider his behavior and emotions about the situations he has experienced | | | | | | |
| | | The student should evaluate the efficiency of the strategies used to regulate his emotions | | | | | | |
| | | The student must adjust the used strategies to regulate their emotions when they are shown to be ineffective | | | | | | |
| | | The student must revise the used methods to deal with emergencies in different situations | | | | | | |

(2)    <u>Teachers' practice of SRL</u>

Your response in this section reflects your practices of self-regulated learning strategies in classrooms. Please choose the most appropriate scale that represents your opinion. (Never) you do not practice it at all; (Always) you practice it.

**Here "supports" include any kinds of both instructional (e.g., lecture, demonstration, modeling, discussion etc.) and noninstructional supports (e.g., rewards, encouragement etc.).**

| S | Items | Never | Rarely | Not Sure | Sometimes | Always |
|---|---|---|---|---|---|---|
| **I Provide my Students with Some Support so that They Can Do the Following Activities by Themselves:** | | | | | | |
| Phase 1: Forethought, planning, and activation | | | | | | |
| 1 | Set their own subgoals for accomplishing the task | | | | | |
| 2 | Think on their own about their prior content knowledge related to the task | | | | | |
| 3 | Think on their own about their past learning experience related to the task | | | | | |
| 4 | Think on their own about the value they can get from accomplishing the task | | | | | |
| 5 | Judge on their own how confident they are for accomplishing the task | | | | | |
| 6 | Think on their own about how much they are interested in the task | | | | | |
| 7 | Plan on their own how they will use time and effort to accomplish the task | | | | | |
| 8 | Plan on their own how they will monitor their learning behavior | | | | | |
| 9 | Think on their own about how they perceive the task | | | | | |
| 10 | Think on their own about how they perceive the study environment | | | | | |
| Phase 2: Monitoring | | | | | | |
| 11 | Self-monitor how well they are learning | | | | | |
| 12 | Self-monitor how motivated they are to accomplish the task or how they feel about their learning | | | | | |
| 13 | Self-monitor their effort, time use, and need for help | | | | | |
| 14 | Self-monitor changes in the task and the study environment conditions | | | | | |
| Phase 3: Control | | | | | | |
| 15 | Use (on their own) cognitive strategies for learning | | | | | |
| 16 | Use (on their own) strategies for managing motivation or affect | | | | | |
| 17 | Decide (on their own) which things to devote more or less effort to | | | | | |
| 18 | Decide (on their own) when, why, and from whom to seek help | | | | | |
| 19 | Change or renegotiate (on their own) the task when needed | | | | | |
| 20 | Change or leave (on their own) the study environment when appropriate | | | | | |
| Phase 4: Reaction and reflection | | | | | | |
| 21 | Self-reflect on how well they did in accomplishing their subgoals | | | | | |
| 22 | Self-reflect on the reasons for their emotional reactions to the outcomes | | | | | |
| 23 | Choose (on their own) if and when to do an additional task | | | | | |
| 24 | Self-evaluate how effective the task was for accomplishing their subgoals | | | | | |
| 25 | Self-evaluate how effective the study environment was | | | | | |

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
