# Peer review of "Impact of General and Special Education Teachers’ Knowledge on Their Practices of Self-Regulated Learning (SRL) in Secondary Schools in Riyadh, Kingdome of Saudi Arabia"

_sustainability, doi:10.3390/su14159420_

Round 1

Reviewer 1 Report

The scientific article "The influence of the knowledge of teachers of general and special education on their practice of self-regulated education (SRL) in secondary schools in Riyadh, Kingdom of Saudi Arabia" touches on the discussion of an important problem and is of some scientific value.

The problem of organizing the practice of self-regulated learning is relevant for teachers (teachers, all employees of the education sector) of any country.

I would like to get acquainted in the article with the description of the most effective methods of organizing self-regulated learning.

Author Response

Dear Reviewer 1

We appreciate the time and effort that you have dedicated to providing your valuable feedback on our manuscript. We are grateful to you for your insightful comments on our paper. We have been able to incorporate changes to reflect your suggestion. We have highlighted the changes within the manuscript.
Here is a  response to your comment:

Comment (1): I would like to get acquainted in the article with the description of the most effective methods of organising self-regulated learning.

Thank you for pointing this out. We agree with this comment. Therefore, we have added a paragraph regarding the ways of promoting self-regulated learning among students. The paragraph highlighted in green colour under a review -Reviewer (1) comment 1- 

Reviewer 2 Report

General note: 

The subject of the work is interesting, but the manuscript must be thoroughly redrafted. The article is repetitive in many places and is not clear in many areas. There are severe flaws in the research methodology—no statistical analysis of the results. The article contains many stylistic, punctuation and grammatical errors. 

Selected detailed comments: 

1. Introduction: The Introduction section must be redrafted to indicate the purpose of the work. Authors must eliminate duplication. Work must follow a logical sequence. 

Lines: 36-51. Please re-edit the passage to eliminate repetition. 

Line 69: "Secondary school" - no need to use capital letters. The more so because the authors use lowercase letters in a different workplace.

 2. Literature review - this part duplicates the information from the Introduction part. Authors should study this section carefully. The reviewer proposes to rewrite this part and add it to the Introduction section. Please pay special attention to removing repetitions, do not duplicate the same information and clearly define the purpose of the work. 

3. Methodology. This part needs to be changed. Authors cannot review the literature again. 

Lines: 258-262. The purpose of the work is duplicated from the Introduction section. 

3.2.1. Participants and Design

 Lines: 279-281. How many teachers? How was the selection of teachers for the study? 

Although the Methodology section is very extensive, finding a methodology for the research isn't easy. Please carefully analyze this section and clearly describe the research methodology, including the statistical analysis of the results. The authors should then discuss the research results and discuss results.

Author Response

Dear Reviewer 2
We appreciate the time and effort that you have dedicated to providing your valuable feedback on our manuscript. We are grateful to you for your insightful comments on our paper. We have been able to incorporate changes to reflect most of the suggestions provided by you. We have highlighted the changes within the manuscript.

Here is a point-by-point response to the reviewers’ comments

Comment (1):Introduction: The Introduction section must be redrafted to indicate the purpose of the work. Authors must eliminate duplication. Work must follow a logical sequence. 

Response: The Introduction section has been redrafted, and the purpose of the work has been presented clearly at the end of 1.1 Background. The section highlighted in yellow colour under a review -Reviewer (2) comment 1-

Comment (2): 36-51. Please re-edit the passage to eliminate repetition.

Response: The whole introduction has been re-edited  

Comment (3): Line 69: "Secondary school" - no need to use capital letters. The more so because the authors use lowercase letters in a different workplace.

Response: We agree with this comment. Therefore, we have changed the capital letter. The word highlighted in yellow colour under a review -Reviewer (2) comment 3-

Comment (4): Literature review - this part duplicates the information from the Introduction part. Authors should study this section carefully. The reviewer proposes to rewrite this part and add it to the Introduction section. Please pay special attention to removing repetitions, do not duplicate the same information and clearly define the purpose of the work.

Response: A literature review has been rewritten and added to the Introduction section. Moreover, we worked to remove repetitions

Comment (5): Methodology: This part needs to be changed. Authors cannot review the literature again

Response: We agree with this comment. Therefore, all parts that discuss literature have been removed from this section if it is repeated or added to the introduction.

Comment (6): 258-262. The purpose of the work is duplicated from the Introduction section

Response: The purpose of the work remained only in the introduction section

Comment (7): 2.1. Participants and Design   Lines: 279-281. How many teachers? How was the selection of teachers for the study? 

Response: Sampling and sample number have been presented. The section highlighted in yellow colour under a review -Reviewer (2) comment 7a- and Reviewer (2) comment 7b

Comment (8): Although the Methodology section is very extensive, finding a methodology for the research isn't easy. Please carefully analyze this section and clearly describe the research methodology, including the statistical analysis of the results. The authors should then discuss the research results and discuss results.

Response: The methodology is described, and the results section has been re-edited

Reviewer 3 Report

The authors quantitatively analyze the impact of gender and knowledge on the self-regulated learning practice of teachers of therapeutic pedagogy or teachers of students with disabilities. A questionnaire is used as a research instrument, on which the authors carry out a very robust validation (based, mainly, on a factor analysis that has allowed the authors to design a theoretical model that explains with statistical significance the answers obtained, on the analysis of the reliability and consistency of the instrument through the computation of the CR and on the convergent validation through the computation of the AVE). The factor analysis of the instrument is correct and robust, and the reliability and convergent validity parameters are adequate, which shows that the instrument used is solid. The authors' analysis of the responses is of a quantitative type based on the comparison of the means, using Student's t-test, of the different scales defined in the model that emerged from the factor analysis, when differentiated by the variables considered (of an academic nature, such as knowledge, or sociological, such as gender). The results obtained conclude that gender does not significantly influence the participants' practice of self-regulated learning, but the more academic variable does.
From my point of view, the object of study is adequate and the methodology used is correct and robustly developed. The article is well structured and clearly presented. I have doubts about the scope of the results. The authors indicate that, among the limitations of the work, is that the random nature of the sample selection detracts a certain potential for the results to be generalizable, although the sample size is acceptable. I agree. However, I believe that the specific characteristics of the educational system and teacher training in Saudi Arabia constitute (or could constitute) the major limitation of the study in terms of generalizability of the results. I would like the authors to include an explanation of the influence of the very specific geographic location of the participants on the conclusions of the study or.
On the other hand, it would be useful for the authors to address the following minor considerations:
- They should adapt the references to the format of the journal.
- Figures 1 and 2 should be adapted so that the text, factorial weights, etc. are more easily readable. These are two very important figures, because they compile the results of the factor analysis carried out on the responses and describe the model that explains these same responses. If the authors improve the quality of the figure, this would be very good.

Author Response

Dear Reviewer 3
We appreciate the time and effort that you have dedicated to providing your valuable feedback on our manuscript. We are grateful to you for your insightful comments on our paper. We have been able to incorporate changes to reflect most of the suggestions provided by you. We have highlighted the changes within the manuscript.

Comment (1): I would like the authors to include an explanation of the influence of the very specific geographic location of the participants on the conclusions of the study  

Response: Thank you for pointing this out. We agree with this comment. Therefore, the requested explanation has been added. The paragraph highlighted in blue color under a review -Reviewer (3) comment 1-

Comment (2): They should adapt the references to the format of the journal.

Response: The references have been adapted to the journal's format 

Comment (3): Figures 1 and 2 should be adapted so that the text, factorial weights, etc. are more easily readable. These are two very important figures because they compile the results of the factor analysis carried out on the responses and describe the model that explains these same responses. If the authors improve the quality of the figure, this would be very good

Response: It is good feedback. However, the authors could not improve the quality of the figures as they copied from the software.

Round 2

Reviewer 2 Report

The work has been corrected according to the reviewer's suggestions. There are still some errors in the work, probably resulting from many changes introduced (please see, for example, the fragment on lines 237-251). The reviewer proposes a thorough check of the entire work.

Author Response

   Dear Reviewer (2) 
Thank you for your valuable feedback. We have been able to incorporate changes to reflect most of the suggestions provided by you. We have highlighted the changes within the manuscript.

Comments and Suggestions for Authors:

The work has been corrected according to the reviewer's suggestions. There are still some errors in the work, probably resulting from many changes introduced (please see, for example, the fragment on lines 237-251). The reviewer proposes a thorough check of the entire work.

Authors' Response:

  • Some parts have been rearranged to maintain consistency between parts of the paper (These parts were highlighted in green)
  • Some parts have been paraphrased (These parts were highlighted in yellow)
